# QTL Mapping for Age-Related Eye Pigmentation in the Pink-Eyed Dilution Castaneus Mutant Mouse

**DOI:** 10.3390/genes13071138

**Published:** 2022-06-24

**Authors:** Takaya Nakano, Momoko Takenaka, Makoto Sugiyama, Akira Ishikawa

**Affiliations:** 1Laboratory of Animal Genetics and Breeding, Graduate School of Bioagricultural Sciences, Nagoya University, Chikusa, Nagoya 464-8601, Japan; nkn.tky91@gmail.com (T.N.); m_takenaka_0@yahoo.co.jp (M.T.); 2Faculty of Veterinary Medicine, Kitasato University School of Veterinary Medicine, Towada 034-8628, Japan; masugi@vmas.kitasato-u.ac.jp

**Keywords:** pink-eyed dilution castaneus, pigmentation, QTL, mice

## Abstract

Pink-eyed dilution castaneus (*Oca2^p-cas^*) is a mutant gene on mouse chromosome 7 that arose spontaneously in wild *Mus* *musculus* *castaneus*. Homozygotes for *Oca2^p-cas^* exhibit pink eyes and a light gray coat throughout life. In an ordinary mutant strain carrying *Oca2^p-cas^*, we previously discovered a novel spontaneous mutation that gradually increases melanin pigmentation in the eyes and coat with aging, and we developed a novel mutant strain that was fixed for the novel phenotype. The purpose of this study was to map major quantitative trait loci (QTLs) for the novel pigmentation phenotype and for expression levels of four important melanogenesis genes, microphthalmia-associated transcription factor (*Mitf*), tyrosinase (*Tyr*), tyrosinase-related protein-1 (*Tyrp1*) and dopachrome tautomerase (*Dct*). We developed 69 DNA markers and created 303 F2 mice from two reciprocal crosses between novel and ordinary mutant strains. The QTL analysis using a selective genotyping strategy revealed a significant QTL for eye pigmentation between 34 and 64 Mb on chromosome 13. This QTL explained approximately 20% of the phenotypic variance. The QTL allele derived from the novel strain increased pigmentation. Although eye pigmentation was positively correlated with *Dct* expression, no expression QTLs were found, suggesting that the pigmentation QTL on chromosome 13 may not be directly in the pathway of any of the four melanogenesis genes. This study is the first step toward identifying a causal gene for the novel spontaneous phenotype in mice and is expected to discover a new regulatory mechanism for complex melanin biosynthesis during aging.

## 1. Introduction

In the mouse, a pilot model animal for humans and livestock, a large number of genetic loci for pigmentation of the coat and eyes have been reported and their information is deposited in Mouse Genome Database (MGD) [1]. At the loci, mutant alleles that control abnormal pigmentation phenotypes are particularly useful not only for elucidating the complex biological mechanisms of melanin pigmentation but also for understanding the etiology of human pigmentation disorders [2]. In general, most mutant genes show congenital phenotypes that do not change throughout life. Uniquely, several mutant genes, including greying with age (*Ga*) [3] and faded (*fe*) [4], have been reported to cause progressive hair graying with aging. Such a hair graying phenotype has also been reported in other mammalian species including horses [5], dogs [6] and humans [7].

In wild mice (*Mus musculus castaneus*), a new coat color mutation, named pink-eyed dilution castaneus (*Oca2^p-cas^*), occurred spontaneously at the *Oca2* locus on chromosome 7 [8]. Mice homozygous for *Oca2^p-cas^* on the C57BL/6JJcl genetic background have pink eyes and a gray coat with no phenotypic changes throughout life [8]. Moreover, in the process of maintaining a mouse strain that was fixed for *Oca2^p-cas^*, we discovered a novel spontaneous mutant that develops black eyes and a darker coat by 3 months of age due to increased pigmentation in ocular choroid and hair follicles [9]. The novel mutant mice have the same deletion of *Oca2* exons 15 and 16 as the ordinary mutant mice with no phenotypic changes throughout life [9]. Furthermore, our previous serum-free primary culture experiments using epidermal cells from neonatal mice showed that the addition of L-tyrosine (Tyr) to the culture medium greatly induces the differentiation of melanocytes from novel mutant mice in a concentration-dependent manner compared to melanocytes from ordinary mutant mice [10]. Immunocytochemical analysis of cultured melanocytes showed that protein expression of tyrosinase (*Tyr*) and tyrosinase-related protein-1 (*Tyrp1*) genes is greatly induced or stimulated in melanocytes of both novel and ordinary mice. However, protein expression of the microphthalmia-associated transcription factor (*Mitf*) gene is stimulated only in melanocytes of novel mice, and furthermore, no difference in protein expression of the dopachrome tautomerase (*Dct*) gene was observed between the two mutant mouse strains [10]. It is well known that *Tyr*, *Tyrp1* and *Dct* genes play important roles in the biosynthesis of two chemically distinct types of melanin, black-brown eumelanin and yellow-brown pheomelanin, and *Mitf* is known to be a master regulator for both eumelanin and pheomelanin production [11]. To the best of our knowledge, no spontaneous mutations that cause progressive darkening of hair and eyes with aging have so far been reported in other mouse strains or in other mammals.

In the present study, we morphologically characterized the novel progressive darkening phenotype in greater detail than in our previous study [9] using novel and ordinary *Oca2^p-cas^* mutant strains and their F1 and F2 progenies. Furthermore, we developed new DNA markers and then performed quantitative trait locus (QTL) analysis for eye pigmentation and eye expression levels of four important melanogenesis genes (*Mitf*, *Tyr*, *Tyrp1* and *Dct*) in reciprocal F2 populations between novel and ordinary mutant strains. Since the difference in pigmentation between the two strains was more pronounced in the eyes than in the hair, we focused on eye pigmentation rather than hair pigmentation as the trait for QTL analysis to map major QTLs for eye pigmentation.

## 2. Materials and Methods

### 2.1. Animals

All animal experiments were performed in accordance with the guidelines for the care and use of laboratory animals of Nagoya University, Japan. The protocol was approved by the Animal Research Committee of Nagoya University.

Ordinary and novel *Oca2^p-cas^* mutant mice have been maintained in our laboratory as B6;Cg-*Oca2^p-cas^*/1Nga (hereafter called PINK) and B6;Cg-*Oca2^p-cas^*/2Nga (BLACK) strains, respectively [9]. For analysis of age-related changes in eye color, PINK(F32-34, 36) and BLACK(F26-31) mice at 1, 2, 3 and 4 months of age were used (*N* = 11–23/age/strain). For QTL analysis, two F2 populations, called PBF2 and BPF2, were produced from reciprocal intercrosses between PINK(F15) and BLACK(F14) mice. The PBF2 population consisted of 164 F2 mice (80 males and 84 females) from an intercross between a PINK female and a BLACK male. The BPF2 population consisted of 139 F2 mice (74 males and 65 females) from an intercross between a BLACK female and a PINK male. All of the mice obtained were raised in an environment with a room temperature of 23 ± 2 °C and a light/dark cycle of 12:12. Commercial chow pellets (CA-1, CLEA Japan, Tokyo) and tap water were given ad libitum.

### 2.2. Whole Genome Sequencing

Genomic DNA was extracted using a DNeasy blood & Tissue kit (Qiagen, Tokyo, Japan) from ear clips of female littermates (*N* = 1 per strain) of the BLACK and PINK parental mice used for F2 populations. The DNA concentration was measured with a Qubit fluorometer (Thermo Fisher Scientific, Tokyo, Japan). Whole genome sequencing with the next-generation sequencer Illumina HiSeq2000 was outsourced to BGI Japan (Kobe, Japan). Sequence reads obtained were mapped to UCSC Mouse Genome Browser NCBI38/mm10 assembly (RefSeq mm10). All read data sequenced were deposited in the DDBJ Sequence Read Archive under the accession number DRA006780.

### 2.3. Stereomicroscopic and Light Microscopic Analyses of Eyes

After taking pictures, mice of parental strains and two F2 populations described above and reciprocal F1 males (*N* = 3 per F2 population) were anesthetized with isoflurane and slaughtered by decapitation. Both eyes of these mice were excised. The right eyes were immediately flash-frozen with liquid nitrogen for RNA extraction and then stored at −80 °C. The left eyes were immediately fixed in 4% paraformaldehyde in 0.1 M phosphate buffer, pH 7.4. After fixation of the eyes for 3 days or more, the eyes were photographed under an Olympus SZX7 Stereomicroscope (Olympus, Tokyo, Japan) equipped with an Olympus DP70 Digital Microscope Camera (Olympus, Tokyo, Japan).

For light microscopic analysis, the fixed eye samples were embedded in paraffin. Sample blocks were sectioned into 4-μm-thick sections, and the sections were deparaffinized through the use of xylene and ethanol series. Serial sections were subjected to hematoxylin-eosin staining for structural examination and Fontana-Masson staining for melanin identification in the retina, as described previously [9]. Slide specimens were observed under an Olympus BX51 System Microscope (Olympus, Tokyo, Japan) equipped with an Olympus DP70 Digital Microscope Camera. The other sections were left unstained to measure brightness in pigmented regions of the retinal pigment epithelium and choroid (Appendix A) as gray values in 256 color gradation obtained by Fiji-ImageJ software version 1.53c (Bethesda, MD, USA) [12].

### 2.4. Light Microscopic Analyses of Coat Hair

Coat hair was pulled out from the dorsal region of each mouse. The hair was wrapped in filter paper and stored at room temperature. The hair samples were dehydrated in a series of ethanol solutions, cleared in ethanol-xylene and xylene, and embedded in Canada balsam, as previously described [9]. Slide specimens were observed under the Olympus BX51 System Microscope.

### 2.5. Real-Time qPCR Analysis

Total RNA was extracted from frozen eyes of mice that exhibited the darkest and lightest eye colors in the PBF2 population by using TRI Reagent (Cosmo Bio, Tokyo, Japan) according to the manufacturer’s instructions. The cDNA was synthesized from 1 μg of total RNA using a PrimeScript^TM^ RT reagent Kit with gDNA Eraser (Takara Bio, Otsu, Japan) according to the manufacturer’s instructions. Quantitative real-time PCR (RT-qPCR) analysis was carried out in a 10.0-µL reaction volume on a StepOnePlus Real-Time PCR system (Thermo Fisher Scientific, Tokyo, Japan) with SYBR Premix Ex Taq™ II (Tli RNaseH Plus) (Takara Bio). Primer sequences for four melanogenesis genes, *Mitf*, *Tyr*, *Tyrp1* and *Dct*, and an endogenous control gene, glyceraldehyde-3-phosphate dehydrogenase (*Gapdh*), are listed in Appendix A. The primers were custom synthesized. The precision of qPCR for the endogenous gene and each of the melanogenesis-related genes was examined by dissociation curves, PCR amplification efficiencies and R^2^ values for quantitative relative standard curves with four serial dilution points of the BLACK mouse cDNA (20 ng, 4 ng, 0.8 ng and 0.16 ng). All samples were analyzed in triplicate. The expression levels of the melanogenesis-related genes were normalized to that of *Gapdh* and measured using the 2^−^^ΔΔCT^ method.

### 2.6. Marker Development

The DNA markers based on SNPs and indels were newly developed to determine the genotypes of F2 mice for QTL analysis. Whole genome sequence data obtained as described above were analyzed by using Integrative Genomics Viewer version 2.11.9 (IGV, https://www.igv.org (accessed on 1 April 2020)) [13] in order to obtain information on SNPs and indels that differ between BLACK and PINK strains. To design PCR primer pairs for SNP markers and indel markers, approximately 3000-bp regions including target SNPs and indels were searched for using Primer3Plus software (https://www.bioinformatics.nl/cgi-bin/primer3plus/primer3plus.cgi (accessed on 1 May 2018)) [14]. The designed primer pairs were checked for non-specific amplification using Primer-Blast software (https://www.ncbi.nlm.nih.gov/tools/primer-blast/ (accessed on 1 May 2018)) [15].

Three types of PCR-based DNA markers were developed and are listed in Appendix A. Two types were SNP markers based on a restriction fragment length polymorphism (RFLP) and a base pair mismatch between one of the primer pairs and its template. For the RFLP marker, an SNP was present within the recognition site of a restriction enzyme that cleaved only one allele and did not cleave the other allele. For the mismatch marker, two pairs of allele-specific primers were designed according to a method previously reported [16]. One primer pair was introduced with an artificial base pair mismatch in the third nucleotide closest to the 3’end of the primer that corresponded to the SNP site, and its target band was not amplified. The other primer pair did not have such a base pair mismatch and its target band was amplified. The last type of DNA markers was indel markers, which amplified different sizes of target bands depending on insertion and deletion.

### 2.7. Genotyping and Linkage Map Construction

For genotyping the DNA markers that were developed, genomic DNA was extracted by a standard method from ear clips of parental strain mice and their F1 and F2 mice. The genotypes of the mice were determined by PCR amplification and agarose gel electrophoresis as previously described [17]. For RFLP markers, after PCR amplification, PCR products were treated with restriction enzymes (Takara Bio and NEB Japan, Tokyo, Japan) (see Appendix A) according to the manufacturer’s instructions. After examining the segregation distortion of the marker genotypes in the F2 population from the expected segregation ratio of 1:2:1 with a chi-square test, a linkage map was constructed using the Kosambi map function of Map Manager QTX b20 software with linkage criterion of *p* = 0.001 [18].

### 2.8. QTL Analysis

Before QTL analysis, the effects of sex, F2 population and their interaction on phenotypic traits were tested using a linear model of JMP Pro version 15.2.1 (SAS Institute Japan, Tokyo, Japan). If the effects were significant at nominal 5% levels, they were included as additive and/or interactive covariates in a model of QTL analysis described below.

To map QTLs cost-effectively, we used a selective genotyping strategy in which only phenotypic extreme individuals were selected from high and low 20–25% of a population and were genotyped for QTL mapping [19]. The QTL analysis was carried out using the simple interval mapping method based on Haley-Knott regression by the function calc.genoprob of R/qtl package version 4.1.2 [20]. To detect QTLs with main effects on a trait, a single-QTL genome scan with a single QTL model was performed by the function scanone of R/qtl. Logarithms of the odds (LOD) scores were calculated at a 1-cM interval across the linkage map constructed. To detect QTLs with additive effects and/or epistatic interaction effects, a two-dimensional genome scan with a two-QTL model was performed by the function scantwo of R/qtl. LOD scores were calculated at a 2.4-cM interval (which was average maker spacing) across the linkage map. Genome-wide significance threshold levels at 0.1%, 5% and 10% levels were computed with 10,000 permutations for scanone and 500 permutations for scantwo.

To find QTLs with context-specific effects on a trait, a single-QTL genome scan was performed for each sex and each population separately, using combined trait data of sex and population. Genome-wide significance threshold levels at 0.1%, 5% and 10% levels for scanone were computed with 10,000 permutations for each sex and each population separately. If QTLs were found at the 10% threshold level in each sex or each population, the statistical significance of QTL-by-sex or QTL-by-population interactions was tested according to the method previously described [20].

For each QTL identified, the percentage of phenotypic variance explained (% Var), additive and dominant effects, and a 1.8-LOD drop (comparable to 95%) confidence interval (CI) were calculated by the function fitqtl of R/qtl. The additive effect was half of the trait difference between two homozygotes for the allele derived from the BLACK strain and the allele derived from the PINK strain. The dominant effect was the difference between heterozygotes for BLACK and PINK alleles and the average between two homozygotes. The degree of dominance (ratio of the dominant effect to the additive effect) was calculated, and it was used to determine the mode of inheritance of the QTL as previously described [21].

### 2.9. Statistical Analysis

Differences in traits among four groups that combined two sexes and two F2 populations were compared by two-way analysis of variance (ANOVA), followed by Tukey’s honestly significant difference (HSD) post hoc test with JMP Pro. Spearman’s rank correlations between phenotypic traits were calculated in each population by JMP Pro.

## 3. Results

### 3.1. Sequencing and Phenotyping of Parental Strains

#### 3.1.1. Sequencing

To determine whether sequence differences in the *Oca2^p-cas^* gene occurred at the time of creation of the F2 populations, whole genome sequencing was performed on two female littermates of BLACK and PINK parental mice, and their alignment statistics are summarized in Appendix A. As shown in Appendix A, the BLACK and PINK females had the same three deletions in the *Oca2* gene. One deletion was 4137 bp in length and contained exons 15 and 16. This deletion was previously reported as a causal mutation of *Oca2^p-cas^* [9]. The other two deletions were newly found in introns 6 and 17 with lengths of approximately 380 bp and 213 bp, respectively. In the *Herc2* gene, which is known as an enhancer to regulate *Oca2* expression [22], the BLACK and PINK females had the same 2919-bp deletion in intron 48 (Appendix A). Comparisons of *Oca2* and *Herc2* sequences with RefSeq mm10 revealed that the two genes in BLACK and PINK strains may have originated in wild *M*. *m*. *castaneus*.

#### 3.1.2. Phenotyping

The appearance of PINK and BLACK mice was examined chronologically at 1, 2, 3 and 4 months of age (*N* = 11–23/age/strain). As is evident in Figure 1a, BLACK mice showed a gradual darkening of eye color from 1 to 3 months of age, while PINK mice did not show any change. This age-related change was validated by observation of changes in eyeball color from 1 to 4 months of age (Figure 1b). Some variation in eyeball coloration was seen in BLACK mice. Light microscopic analysis of hematoxylin-eosin-stained and Fontana-Masson-stained eyes revealed that the choroid of a BLACK male at 9 months of age was more heavily pigmented than that of a PINK female at 11 months of age and that their F1 mouse showed intermediate pigmentation between the two parental mice (Appendix A).

### 3.2. Phenotyping of F2 Mice

#### 3.2.1. Eye Morphology

The eye color of both PBF2 and BPF2 mice gradually changed from light to dark, as easily observed by the naked eye (Figure 2). PBF2 mice showed a bimodal distribution of eye color. To perform selective genotyping, 26–28% of the darkest and lightest individuals were selected from the PBF2 population (Figure 2a), while 24% of those individuals were selected from the BPF2 population (Figure 2b). Light microscopy of hematoxylin-eosin-stained and Fontana-Masson-stained eyes in the PBF2 population revealed that the retinal pigment epithelium and choroid of black-eyed mice were more heavily pigmented than those of pink-eyed mice at 4 months of age (Figure 3). In contrast, age-related changes in dorsal awl hair color from 3 weeks to 4 months of age appeared to be less clear in both pink-eyed and black-eyed mice (Appendix A).

Mice with abnormal eye morphology occasionally appeared in the PBF2 and BPF2 populations. Eye abnormalities in 17 mice in the PBF2 population were morphologically classified into four types: cataract, microphthalmia, anophthalmia and eyelid coloboma (Appendix A and Table 1). These abnormalities were also seen in two parental strains of PINK and BLACK.

#### 3.2.2. Traits

For QTL analysis, traits of two pigmentations (binary and gray values) and four gene expression levels (*Mitf*, *Tyr*, *Dct* and *Tyrp1*) were measured in eyes of selected individuals in the PBF2 and BPF2 populations (see Figure 2 for the numbers of selected individuals) and their measurements are summarized in Table 2. The binary value was obtained by scoring the darkest individuals as 1 and the lightest individuals as 0 by the naked eye. The *Tyr* expression level was significantly different between the two populations and also between males and females, but the population-by-sex interaction was not significant. The *Tyrp1* expression level showed a significant difference only between the two populations. The other traits did not show any significant differences.

As shown in Appendix A, Spearman’s rank correlation analysis revealed that the binary value showed a highly significant positive correlation with the gray value. Both binary and gray values were positively correlated with *Dct* expression levels only, but the four gene expression levels were positively correlated with each other at the 5% level after Bonferroni correction.

### 3.3. Marker Development and Linkake Map Construction

Whole genome sequencing identified 243,411 and 145,937 SNPs specific to BLACK and PINK strains, respectively. Similarly, 68,409 and 43,069 indel variants were detected in BLACK and PINK strains, respectively. Based on the SNP and indel information, 110 PCR primer pairs were designed to develop RFLP, mismatch and indel markers. Genotyping of BLACK, PINK and their F1 mice showed that 73 of the 110 markers were informative between BLACK and PINK strains. Among the 73 markers, 69 statistically met the expected segregation ratio of 1:2:1 at the nominal 5% level in the 88 individuals selected from the PBF2 population. Finally, at least two of the 69 markers were placed on each chromosome to cover all autosomes and the X chromosome. Details of the marker information, such as primer sequences and PCR conditions, are shown in Appendix A.

After genotyping 88 and 67 individuals selected from the PBF2 and BPF2 populations, respectively, three genetic linkage maps for the 69 markers were constructed with Map Manager QTX software, as shown in Figure 4. Summary statistics for the linkage maps are shown in Appendix A. In the PBF2 population, 64 of the 69 markers were assigned to 18 linkage groups on 19 autosomes and the X chromosome with an average marker spacing of 14.3 cM (Appendix A). In the BPF2 population, 54 markers were assigned to 23 linkage groups on 19 autosomes and the X chromosome with an average marker spacing of 14.5 cM (Appendix A). In the combined F2 population, 62 markers were assigned to 17 linkage groups on 19 autosomes and the X chromosome with an average marker spacing of 14.9 cM (Appendix A). The total length of the linkage maps varied from 608.1 to 672.1 cM.

### 3.4. QTL Analysis

#### 3.4.1. Main-Effect QTL

To detect QTLs with main effects on traits of eye pigmentation and gene expression levels, a single-QTL genome scan using the linkage map constructed above was performed with R/qtl software in each of the three F2 populations. Genome-wide LOD score plots for all traits are shown in Appendix A. No QTLs with main effects on gene expression levels were identified at genome-wide 10% threshold levels, except for eye pigmentation for binary and gray values described below.

As shown in Figure 5a and Table 3, in the PBF2 population, a significant QTL with a main effect on binary values with a peak LOD score of 4.5 was identified at 8.8 cM (54.6 Mb) on chromosome 13 with a 95% CI of 34–64 Mb, at the genome-wide 5% threshold level. At the same map position, a significant QTL affecting gray values (LOD = 4.6) was found. The LOD scores for both values changed in almost the same pattern across chromosome 13, with a slightly higher LOD score for gray values than that for binary values (Figure 5a). The QTLs for binary and gray values explained about 21% of the corresponding phenotypic variance, and their alleles derived from the BLACK strain increased pigmentation in a dominant fashion (Table 3 and Appendix A).

The nearest marker, *13_54*, to the detected QTL on chromosome 13 was an indel marker (Appendix A). The forward and reverse primers for this marker were located in intron 2 and exon 4, respectively, of the ADP-ribosylation factor-like 10 (*Arl10*) gene. At this marker locus, the PCR product size of the allele derived from the PINK strain was 1770 bp, matching with that of the allele derived from the reference C57BL/6J strain. In contrast, the product size of the allele derived from the BLACK strain was 970 bp due to a deletion that appears to include part of *Arl10* exon 4.

No QTLs for binary values were found in the BPF2 population (Figure 5b). However, a highly significant QTL for this trait (LOD = 5.2) was identified at 9.0 cM (54.8 Mb) on chromosome 13 at the genome-wide 0.1% threshold level in the combined F2 population (Figure 5c). This QTL explained 14.3% of the phenotypic variance, and the BLACK-derived allele increased pigmentation in an additive fashion (Table 3). In both PBF2 and combined F2 populations, an additional small LOD peak was seen at approximately 3 cM (Figure 5a,c). However, a two-dimensional genome scan did not provide statistical evidence for the presence of the additional QTL.

#### 3.4.2. Context-Specific QTL

As described in 3.2.2, *Tyr* and *Tyrp1* expression levels showed a significant difference between PBF2 and BPF2 populations (Table 2). To determine whether population-specific QTLs are present, a single-QTL genome scan was performed in each population separately. However, no population-specific QTLs were identified at genome-wide 10% threshold levels (Appendix A). *Tyrp1* expression showed a significant sex difference (Table 2), but no sex-specific QTLs were found at the genome-wide 10% threshold level (Appendix A).

## 4. Discussion

For QTL mapping, we newly developed 69 DNA markers that were located on 19 autosomes and the X chromosome with an average marker spacing of 14–15 cM. This spacing is not a very small spacing because a marker spacing of 30 cM is generally known to be optimal for initial QTL mapping using an F2 population [23]. Furthermore, QTL analyses in backcrosses and F2 populations generally localize a QTL to a large genomic interval of 10–50 cM, as previously reviewed [24]. Thus, the number of markers developed in this study is by no means small for mapping major QTLs in the F2 population. However, we cannot rule out the possibility that QTLs were missed on chromosomal regions not covered by the markers developed.

We used two strategies for QTL mapping to reduce the cost and effort of phenotyping and genotyping as much as possible. One strategy was to use binary values as an easy-to-measure trait. Its validity was verified in the PBF2 population by our two results: the high positive correlation between the binary trait and microscopically measured gray values and the similarity of parameter estimates for the two QTLs affecting binary and gray values. The two results clearly indicate that the two QTLs are the same locus. The failure to detect a binary QTL in the BPF2 population might be due to two possible reasons. One is the smaller sample size of the BPF2 population, whose QTL was found in PBF2 and combined F2 populations. The other is that the BPF2 mice had lower eye color contrast than the PBF2 mice. The other strategy was to use selective genotyping. We selected 24–28% of the extreme individuals from each of the PBF2 and BPF2 populations. This percentage of selected individuals is considered to be sufficient for QTL mapping because it has been reported that selecting 20–25% in each tail of a phenotypic distribution is sufficient to accurately map a QTL and that selecting more than 20–25% does not reduce the length of the 95% CI [25].

The chromosome 13 QTL found in the present study explained approximately 20% of the phenotypic variance. This proportion may have been overestimated by the Beavis effect [26] due to the small size of the F2 mice used in QTL mapping. Recently, using an integrated approach of gene expression, congenic/subcongenic strain analysis, quantitative complementation testing and causal analysis, we successfully identified the lymphocyte antigen 75 (*Ly75*) gene as the causal gene for *Pbwg1.5*, a QTL that affects resistance to obesity in mice [27]. The additive effect of *Pbwg1.5* was reported to be 0.38 in standard deviation units and the dominant effect was 0.03 [17], suggesting that the phenotypic effect of *Pbwg1.5* is small. On the other hand, when the measurements obtained in the present study were converted to units of standard deviation, the additive effect of the pigmentation QTL for binary and gray values in the PBF2 population ranged from 0.54 to 0.58, and the dominant effect ranged from 0.32 to 0.38. This indicates that the phenotypic effect of the pigmentation QTL is greater than that of *Pbwg1.5*. Thus, future genetic analysis using congenic and subcongenic strains may confirm the pigmentation QTL and further identify the causal gene.

In the present study, we found no sequence differences between BLACK and PINK strains for *Oca2^p-cas^* and its enhancer *Herc2* on chromosome 7. Furthermore, no expression QTLs were found for *Mitf*, *Tyr*, *Tyrp1* and *Dct* genes on chromosomes 6, 7, 4 and 14, respectively, although there was a positive correlation between *Dct* expression and eye pigmentation (binary and gray values). These results clearly indicate that none of the six genes is responsible for the novel mutant phenotype, suggesting that the pigmentation QTL on chromosome 13 may not be directly in the pathway of any of the six genes.

The *13_54* marker nearest to the chromosome 13 QTL was located on the *Arl10* gene. The BLACK-derived allele at this marker locus had a deletion that appears to include part of *Arl10* exon 4. According to MGD (last database update 5 March /2022) [1], mice knocked out for *Arl10* show decreased bone mineral content, decreased lean body mass and increase total body fat amount. In the PBF2 population, body weight was measured at 4 months of age, but bone mineral content and body fat mass were not measured. We performed QTL analysis for body weight, but no QTLs were identified.

Using MGD [1], we searched for protein-coding genes in the 95% CI (34–64 Mb) of the chromosome 13 QTL and obtained a list of five genes for pigmentation and eye morphology. These genes are biogenesis of lysosomal organelles complex-1, subunit 5, muted (*Bloc1s5*), transcription factor AP-2, α (*Tfap2a*), dystrobrevin-binding protein 1 (*Dtnbp1*), msh homeobox 2 (*Msx2*) and growth arrest-specific 1 (*Gas1*). Mutations in *Bloc1s5* [28,29] and *Dtnbp1* [30,31] have been reported to cause pigment dilution in both eyes and coat, prolonged bleeding time and inner ear abnormalities. The human homologs, *BLOC1S5* and *DTNBP1*, are known to be causal genes for Hermansky-Pudlak Syndromic albinism types 11 and 7, respectively [32]. In normal mouse melanocytes, it has been reported that MITF and TFAP2A proteins bind together to the enhancer element in the intron 4 of interferon regulatory factor (4*Irf4*) and cooperatively regulate *Irf4* expression, resulting in activation of *Tyr* expression and thus normal pigmentation [33]. Conversely, disruption of TFAP2A binding decreases *Irf4* expression and then pigmentation. Mice knocked out for *Msx2* exhibit cornea-lentoid adhesions and microphthalmia due to a developmental failure of the lens, a phenotype similar to human Peters anomaly [34]. In addition, mice knocked out for *Gas1* exhibit microphthalmia in which the ventral retinal pigmented epithelium is overproliferated and converted into the neural retina [35].

In aged hairless mice, localized epidermal hyperpigmentation is reported on the trunk, even in the absence of UV irradiation, and it becomes darker with age [36], similar to melasma or lentigines in human skin [37]. As previously reviewed [38], haploinsufficiency in the a disintegrin and metallopeptidase domain 10 (*Adam10*) gene causes freckle-like patches in hairless mice. These mice not only exhibit freckle-like pigmentation on the dorsal aspect of the forelegs, but also diffuse pigmentation on the trunk in adults, a phenotype that serves as a mouse model for the reticulate acropigmentation of Kitamura disease in humans. Notably, haploinsufficiency of *Adam10* alone, without homozygous hairless mutations, does not cause this pigmentation. In addition to the hairless mice, the present novel *Oca2^p-cas^* mutant may provide a useful animal model for elucidating the mechanisms of age-related pigmentation in humans and other mammals.

In conclusion, this study is the first step toward identifying a causal gene for the chromosome 13 QTL affecting age-related pigmentation in our novel spontaneous mouse mutant. This causal gene may not be directly in the pathway of any of four important melanogenesis genes (*Mitf*, *Tyr*, *Tyrp1* and *Dct*). We are now developing a congenic strain carrying the QTL interval on the genetic background of the PINK strain. Identification of the causal gene will lead to the discovery of a new regulatory mechanism for age-related melanin biosynthesis.

## Figures and Tables

**Figure 1 genes-13-01138-f001:**
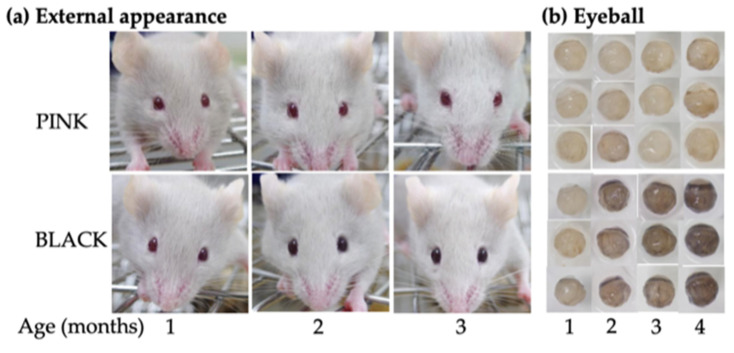
Age-related changes in eye color of ordinary pink-eyed (PINK) and novel black-eyed (BLACK) mutant strains on the C57BL/6JJcl background: (**a**) External appearances of a PINK mouse at the F36 generation and a BLACK mouse at the F31 generation. The same mouse in each strain was photographed from 1 to 3 months of age; (**b**) stereomicrographs of eyeballs from PINK mice at F32–33 generations and BLACK mice at F26–29 generations.

**Figure 2 genes-13-01138-f002:**
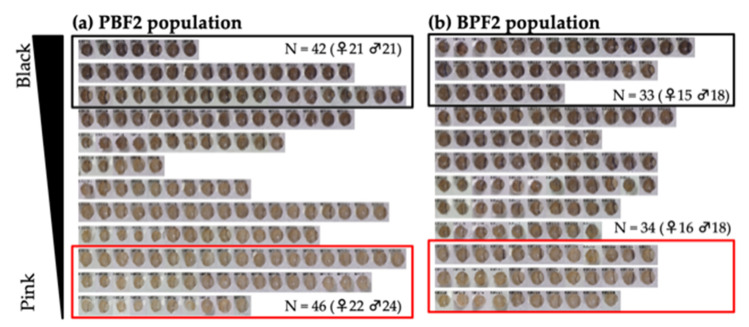
Distribution of eye color in F2 mice at 4 months of age: (**a**) 164 mice in the PBF2 population obtained from an intercross between a PINK female and a BLACK male; (**b**) 139 mice in the BPF2 population obtained from an intercross between a BLACK female and a PINK male. Approximately 25% of the mice with the darkest eye color surrounded by a black border and the lightest eye color surrounded by a red border were selected from each F2 population. *N* indicates the number of mice selected.

**Figure 3 genes-13-01138-f003:**
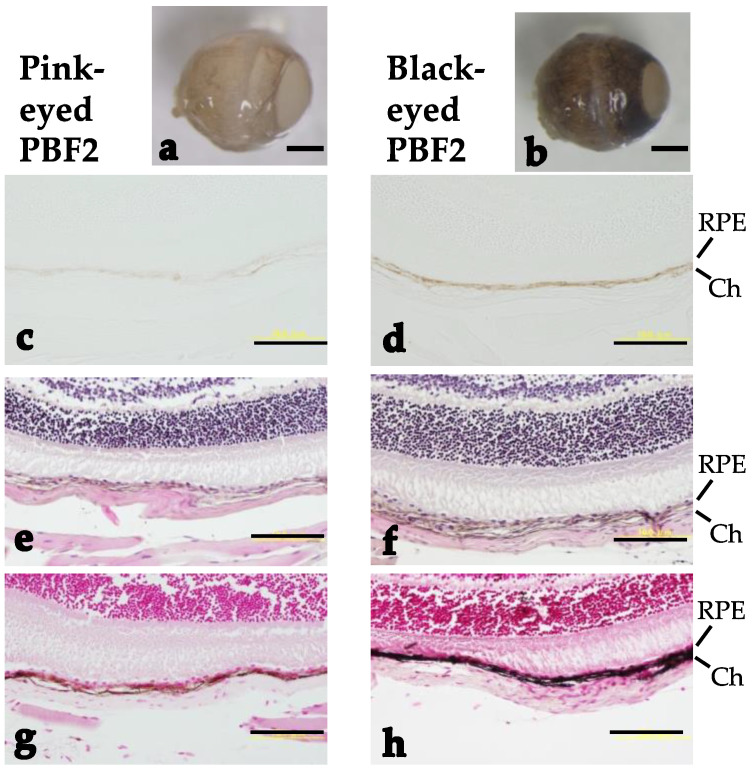
Stereomicroscopic and light-microscopic analyses of eyes from pink-eyed and black-eyed PBF2 male mice at 4 months of age: (**a**,**b**) Stereomicrographs; (**c**,**d**) unstained micrographs; (**e**,**f**) hematoxylin-eosin-stained micrographs; (**g**,**h**) Fontana-Masson-stained micrographs. RPE and Ch indicate retinal pigment epithelium and choroid, respectively. Scale bars show (**a**,**b**) 1 mm and (**c**–**h**) 100 μm.

**Figure 4 genes-13-01138-f004:**
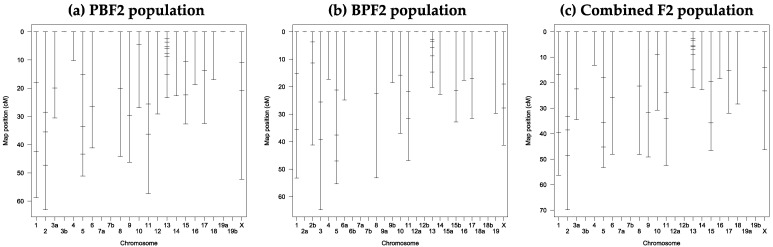
Genetic linkage maps of 69 DNA markers developed in this study: (**a**) PBF2 population; (**b**) BPF2 population; (**c**) combined F2 population. See Appendix A for details of the markers and linkage groups constructed. The small horizontal line on the linkage group indicates the position of the DNA marker.

**Figure 5 genes-13-01138-f005:**
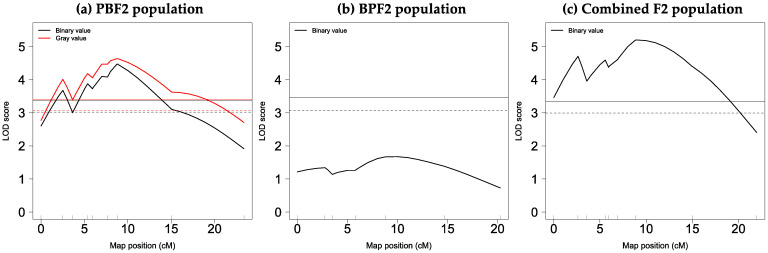
LOD score plots for binary and gray values on chromosome 13: (**a**) PBF2 population; (**b**) BPF2 population; (**c**) combined F2 population. The solid and dashed horizontal lines show genome-wide 5% and 10% significant threshold levels obtained by 10,000 permutations for each trait (see Appendix A). The small vertical line on the *x*-axis indicates the position of each DNA marker.

**Table 1 genes-13-01138-t001:** Number of eye abnormalities observed in 17 mice of the PBF2 population.

Abnormality	Male			Female			Total No. of Cases
	Left Eye	Right Eye	Both Eyes	Left Eye	Right Eye	Both Eyes	
Cataract	1	0	0	5	3	0	9
Microphthalmia	0	1	0	2	0	1	4
Anophthalmia	0	0	0	0	4	1	5
Eyelid coloboma	0	0	0	3	1	0	4
Total no. of cases	1	1	0	10	8	2	22

Five of the 17 mice had two abnormalities, each of which was counted separately.

**Table 2 genes-13-01138-t002:** Trait measurements in PBF2 and BPF2 populations and the effects of population and sex on the measurements.

	PBF2		BPF2		*p*-Value		
Trait	Male	Female	Male	Female	Population	Sex	Population × Sex
Binary value	0.47 ± 0.08	0.48 ± 0.08	0.50 ± 0.08	0.48 ± 0.09	0.86	0.97	0.82
Gray value	180.8 ± 5.2	179.4 ± 5.3	NA	NA	NA	0.85	NA
*Mitf* expression	1.17 ± 0.05	1.23 ± 0.06	1.15 ± 0.06	1.10 ± 0.06	0.19	0.92	0.41
*Tyr* expression	1.30 ± 0.06 ^a,b^	1.48 ± 0.06 ^a^	1.12 ± 0.06 ^b^	1.21 ± 0.07 ^b^	0.00044	0.039	0.50
*Dct* expression	1.53 ± 0.07	1.62 ± 0.08	1.69 ± 0.08	1.62 ± 0.09	0.32	0.91	0.32
*Tyrp1* expression	1.49 ± 0.06 ^a,b^	1.56 ± 0.07 ^a^	1.20 ± 0.07 ^c^	1.26 ± 0.08 ^b,c^	0.000038	0.33	0.97

*p*-values were obtained by two-way ANOVA; ^a–c^ Least squares means ± SEM with different superscript letters within a trait indicate significant differences among four groups combined for sex and population at *p* < 0.05 (Tukey’s HSD test); NA, not applicable.

**Table 3 genes-13-01138-t003:** Summary of QTLs for eye pigmentation on chromosome 13 identified in this study.

Population	Trait	Position ^1^	LOD	% Var	Nearest Marker	CI ^2^	Additive Effect ^3^	Dominant Effect ^3^	d/a	Inheritance
PBF2	Binary value	8.8 (54.6)	4.5 **	20.8	*13_54*	0–20.0 (34–64)	0.27 ± 0.05	0.19 ± 0.10	0.70	Dominant
PBF2	Gray value	8.8 (54.6)	4.6 **	21.8	*13_54*	0–23.0 (34–64)	19.90 ± 4.31	11.10 ± 6.71	0.56	Dominant
Combined F2	Binary value	9.0 (54.8)	5.2 ***	14.3	*13_54*	0–19.0 (34–64)	0.25 ± 0.05	0.06 ± 0.08	0.24	Additive

^1^ Linkage position in cM and physical position in Mb (based on RefSeq mm10) in parenthesis; ^2^ 95% confidence interval (CI) in cM and physical interval in Mb in parenthesis; ^3^ The positive sign of the additive and dominant effects (mean ± SEM) indicates that the allele derived from the BLACK strain increased the trait value; a/d, Degree of dominance that is a ratio of the dominant effect to the additive effect; ***, ** Significant at genome-wide 0.1% and 5% levels, respectively (see Appendix A for LOD threshold levels).

## Data Availability

All data obtained in this study are included in this publication and Appendix A. The data for whole genome sequencing were deposited in the DDBJ Sequence Read Archive under the accession number DRA006780.

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
