# Peer review of "QTL Mapping for Age-Related Eye Pigmentation in the Pink-Eyed Dilution Castaneus Mutant Mouse"

_genes, 2022, doi:10.3390/genes13071138_

Round 1

Reviewer 1 Report

Dear authors, my major concern regarding the present study is the methodology used for QTL analysis. You performed whole genome sequencing in two specimens and detected many thousands of SNP markers and indels. Why did you use only 69 markers for linkage mapping?? In my opinion, the constructed maps are of very low density. 

You also used selective genotyping in two populations for QTL analysis. Selective genotyping is a widely-used strategy and is proved to be very efficient in QTL studies. However, the number of animals genotyped herein (88 in PBF2 population and 67 in BPF2 population) is quite low...

So, from my point of view, the low number of markers combined with the low number of animals can reduce dramatically the statistical power for QTL detection. That's why the QTL for binary and gray values was detected only in PBF2 population and could not be confirmed in BPF2 population. You could also have missed QTLs of low and medium effect!

In addition, the low number of markers and animals used in QTL studies tend to overestimate the effect of the detected QTL. So, the proportion of phenotypic variance (PVE) found herein (20%) might be an overestimation and you should make a comment on that.

The phenotyping and gene expression methodology used herein are fine. I have no comment on that.

The nearest marker (13_54) to the detected QTL on chromosome 13 is a SNP marker or an indel?? Did you perform an association analysis between the different genotypes of this marker and the phenotypes of animals to check for possible statistical significance? Maybe there is a specific genotype of this marker which corresponds to a specific phenotype. In my opinion, you should perform this type of analysis.

Regarding Discussion section, you could be a little bit more descriptive...The discussion is a bit short for this type of study. I suggest you add more comments/speculations/assumptions regarding the interpretation of your results, as well as, discussing (in more detail) your findings in contrast to other QTL studies...

Author Response

Replies to Reviewer 1

We thank Reviewer 1 for your helpful and constructive comments on the manuscript.

Dear authors, my major concern regarding the present study is the methodology used for QTL analysis. You performed whole genome sequencing in two specimens and detected many thousands of SNP markers and indels. Why did you use only 69 markers for linkage mapping?? In my opinion, the constructed maps are of very low density.

Answer: As described in Introduction (Lines 42-46), the novel mutant arose spontaneously in the process of maintaining the ordinary strain. So, initially, before whole genome sequencing, was done, we expected that the number of genetic variants between the novel and ordinary strains would be small. Surprisingly, the study identified a large number of genetics variants. We then came up with the idea of using the sequence information obtained to develop DNA markers. As described in the first paragraph of Discussion, the markers developed have an average marker interval of 14-15 cM. Furthermore, QTL analyses in mouse backcrosses and F2 populations generally localize a QTL to a large genomic interval of 10-50 cM, as previously reviewed [24]. Thus, the number of markers developed in this study is by no means small for mapping major QTLs in the F2 population. Of course, QTLs on chromosomal regions not covered by the markers could be missed. We have added the above sentences to the first paragraph of Discussion.

You also used selective genotyping in two populations for QTL analysis. Selective genotyping is a widely-used strategy and is proved to be very efficient in QTL studies. However, the number of animals genotyped herein (88 in PBF2 population and 67 in BPF2 population) is quite low...

Answer: We selected 88 mice in the PBF2 population and 67 mice in the BPF2 population. However, in total, we developed 164 mice in the PBF2 population, 139 mice in the BPF2 population and 303 mice in the combined population. Initially, we wanted to increase the total number of F2 mice and selected mice, but making light microscopic specimens of eyes and measuring gray values of sections were hard work. We now believe that the number of mice selected is adequate for the initial QTL mapping. In the near future, the presence of the chromosome 13 QTL will be confirmed by congenic/subcongenic mouse analyses carrying the QTL.

So, from my point of view, the low number of markers combined with the low number of animals can reduce dramatically the statistical power for QTL detection. That's why the QTL for binary and gray values was detected only in PBF2 population and could not be confirmed in BPF2 population. You could also have missed QTLs of low and medium effect!

Answer: The number of markers used for QTL mapping was the same between PBF2 and BPF2 populations, but the number of mice used was smaller in BPF2 than in PBF2. The small number is a possible reason for not being able to confirm the QTL in BPF2, as stated by the reviewer. We consider that another possible reason is that the eye color contrast of F2 mice in the BPF2 population was lower than in the PBF2 population (Figure 2). A stated by the reviewer, we were unable to find any QTLs with low or moderate effects in this study. Identification of such QTL requires more than doubling the number of F2 mice, which is a challenge for the future. These sentences have been added to the Discussion section of the revised manuscript.

In addition, the low number of markers and animals used in QTL studies tend to overestimate the effect of the detected QTL. So, the proportion of phenotypic variance (PVE) found herein (20%) might be an overestimation and you should make a comment on that.

Answer: We agreed with the comment of the reviewer. The proportion of variance explained by the QTL may be overestimated by the Beavis effect [26]. We have added the sentence to Discussion.

The phenotyping and gene expression methodology used herein are fine. I have no comment on that.

Answer: Thank you.

The nearest marker (13_54) to the detected QTL on chromosome 13 is a SNP marker or an indel?? Did you perform an association analysis between the different genotypes of this marker and the phenotypes of animals to check for possible statistical significance? Maybe there is a specific genotype of this marker which corresponds to a specific phenotype. In my opinion, you should perform this type of analysis.

Answer: The 13_54 marker is an indel marker (Table S2). The forward and reverse primers for this marker are located in intron 2 and exon 4, respectively, of the ADP-ribosylation factor-like 10 (Arl10) gene. At this marker locus, the PCR product size of the allele derived from the PINK strain was 1770 bp, matching with that of the allele derived from the reference C57BL/6J strain. In contrast, the product size of the allele derived from the BLACK strain was 970 bp due to a deletion that appears to include part of Arl10 exon 4. According to MGD, Arl10 knockout mice show decreased bone mineral content, decreased lean body mass and increase total body fat amount. In the PBF2 population, body weight was measured at 4 months of age, but bone mineral content and body fat mass were not measured. We performed QTL analysis for body weight, but no QTLs were identified. These sentences have been added to Results and Discussion sections.

Regarding Discussion section, you could be a little bit more descriptive...The discussion is a bit short for this type of study. I suggest you add more comments/speculations/assumptions regarding the interpretation of your results, as well as, discussing (in more detail) your findings in contrast to other QTL studies...

Answer: In addition to our text answered in the comments above, we have added some discussion, etc. to the Discussion section.

Reviewer 2 Report

Paper entitled “QTL Mapping for Age-related Eye Pigmentation in the Pink-eyed Dilution Castaneus Mutant Mouse” is quite interesting. The authors provide new information about the QTLs potentially related to Eye Pigmentation in the mouse. However, I didn’t see any information about the implementation of observed results in practice. It is very important to add a clear message for scientists and mainly other readers (introduction/discussion/conclusion) about how this type of study/results can be used/beneficial for humans or livestock. In its current state, it just provides comprehensive information about methodology and obtained results without a real connection to knowledge transfer to the mammalian model. It will be useful to explain the function of genes/QTLs mentioned in the main text in other mammalian species.

Minor points:

L21: The proportion of variability described by QTL is relatively low to reliable confirm its effect. Add information about other factors affecting the variability of the tested trait.

L341-348: Why the sentence fragments are coloured in blue? Revise

Author Response

Replies to Reviewer 2

We thank Reviewer 2 for your helpful and constructive comments on the manuscript.

Paper entitled “QTL Mapping for Age-related Eye Pigmentation in the Pink-eyed Dilution Castaneus Mutant Mouse” is quite interesting. The authors provide new information about the QTLs potentially related to Eye Pigmentation in the mouse. However, I didn’t see any information about the implementation of observed results in practice. It is very important to add a clear message for scientists and mainly other readers (introduction/discussion/conclusion) about how this type of study/results can be used/beneficial for humans or livestock. In its current state, it just provides comprehensive information about methodology and obtained results without a real connection to knowledge transfer to the mammalian model. It will be useful to explain the function of genes/QTLs mentioned in the main text in other mammalian species.

Answer: Thank you for your constructive comments. In the revised manuscript, some text has been added.

In aged hairless mice, localized epidermal hyperpigmentation is reported on the trunk, even in the absence of UV irradiation, and it becomes darker with age [37], similar to melasma or lentigines in human skin [38]. As previously reviewed [39], haploinsufficiency in the a disintegrin and metallopeptidase domain 10 (Adam10) gene has been reported to cause freckle-like patches in hairless mice. These mice not only exhibit freckle-like pigmentation on the dorsal aspect of the forelegs, but also diffuse pigmentation on the trunk in adults, a phenotype that serves as a mouse model for the reticulate acropigmentation of Kitamura disorder in humans. Notably, Adam10 haploinsufficiency alone, without the homozygous hairless mutation, does not cause this pigmentation. In addition to the hairless mice, the present novel mutant may become a useful animal model for elucidating the mechanisms of age-related pigmentation in humans and other mammals.

Minor points:

L21: The proportion of variability described by QTL is relatively low to reliable confirm its effect. Add information about other factors affecting the variability of the tested trait.

Answer: As we responded in our comments to Reviewer 1, the proportion of variance explained by the pigmentation QTL is probably overestimated by the Beavis effect [26]. Recently, using an integrated approach of gene expression, congenic/subcongenic strain analysis, quantitative complementation testing and causal analysis, we successfully identified the Ly75 gene as the causal gene for Pbwg1.5, a QTL that affects resistance to obesity in mice [27]. The additive effect of the Pbwg1.5 QTL was 0.38 in standard deviation units and the dominant effect was 0.03 [28], suggesting that the phenotypic effect of this QTL is small. In contrast, the additive effect of the pigmentation QTL for binary and gray values in the PBF2 population ranged from 0.54 to 0.58 in standard deviation units, and the dominant effect ranged from 0.32 to 0.38. This indicates that the phenotypic effect of the pigmentation QTL is greater than that of Pbwg1.5. Thus, we believe that future genetic analysis using congenic and subcongenic strains may confirm the pigmentation QTL and further identify the causal gene. We added these discussions to the Discussion section of the revised manuscript.

L341-348: Why the sentence fragments are coloured in blue? Revise

Answer: We are sorry. The blue color was removed.

Round 2

Reviewer 1 Report

All comments have been answered by the authors. No need to re-review.

Reviewer 2 Report

The authors added short information about the potential importance of observed results for elucidating the mechanisms of age-related pigmentation in humans and other mammals. Even if the information is really short and doesn’t fully meet my expectation related to the practical use of results, I recommend accepting the paper for publication.